# Passability of Chironomid Larvae in Granular Activated Carbon

**DOI:** 10.3390/ijerph19021005

**Published:** 2022-01-17

**Authors:** Cha Young Lee, Jinseok Byeon, Min Kyung Kim, Tae-gwan Lee, Dong Gun Kim

**Affiliations:** 1Institute of Environmental Ecology, Sahmyook University, Seoul 01795, Korea; sincea@naver.com (C.Y.L.); min4190@naver.com (M.K.K.); 2Department of Chemistry & Life Science, Sahmyook University, Seoul 01795, Korea; a213079@naver.com; 3Department of Bio & Environment Technology, Seoul Women’s University, Seoul 01797, Korea; 4Department of Environmental Science, Keimyung University, Daegu 42403, Korea; wateree@kmu.ac.kr; 5Smith College of Liberal Arts, Sahmyook University, Seoul 01795, Korea

**Keywords:** *Glyptotendipes tokunagai*, granular activated carbon, passability, filtration, backwashing

## Abstract

We conducted an experiment to confirm the passability of chironomid larvae (*Glyptotendipes tokunagai*) in granular activated carbon (GAC) used in water treatment plants. After injecting larvae at different growth stages (first through fourth instars) into circular columns filled with GAC, the number of individuals and their locations within the GAC columns were recorded after 168 h. We found that more than 80% of the injected larvae in the first instar and 3.3% in the second instar passed, whereas none from the third and fourth instars had passed through the column. The second instar larvae were evenly distributed within the column, whereas the third and fourth instar larvae were mostly distributed within 10 cm of the upper layer of the GAC. Our results demonstrate the passability of chironomid larvae in GAC and can be used as basic information for water quality management in water treatment plants.

## 1. Introduction

Supplying high-quality drinking water is very important for human welfare. Supply standards of drinking water include the absence of pathogenic microorganisms, inorganic and organic substances that have a detrimental effect on health, substances that affect the aesthetics of drinking water, and other substances that can have harmful effects on human health [1]. These contaminants are present in rivers, lakes, groundwater, and reservoirs used as drinking water resources [2]. To remove these contaminants, water from a reservoir is usually purified through coagulation with chemical additions, sedimentation, and filtration. There are many different types of materials used in filters, and granular activated carbon (GAC), which has a high adsorption capacity for these substances, is widely used to remove them in the filtration processes of water treatment plants [3,4].

In Korea, the first reported occurrence of macroinvertebrates in domestic tap water supplied from water treatment plants occurred in Incheon in July 2020. Macroinvertebrates were identified as chironomid larvae, and as a result of a nation-wide inspection of water treatment plants, larvae were detected at 7 of 49 sites [5]. The family Chironomidae is the most dominant and cosmopolitan family in the order Diptera [6]. Because chironomid larvae are resistant to pollution, they have been used as test species for the risk assessment of various contaminants [7]. The larvae found in the water treatment plants in the Incheon area were identified as *Chironomus kiiensis*, *Chironomus flaviplumus*, *Chironomus dorsalis*, and *Polypedilum yongsanensis* [8]. If the infiltration route of chironomid adults into water treatment plants is not completely blocked, it is possible that these insects may multiply.

The assessment of drinking water quality is accomplished based on physicochemical criteria. The problems associated with occurrence of invertebrates in drinking water have received increasing attention in recent years. The occurrence of visible invertebrates in tap water causes psychological discomfort to consumers and decreases consumer confidence in tap water quality [8,9,10,11,12,13,14,15,16]. In addition, it can lead to indirect negative effects such as microbial regrowth [11,12]. In the case of benthic macroinvertebrates, most of these are eliminated during the filtration of the surface water, but it has been reported that some remain in the GAC or biological activated carbon (BAC) filter layer and can survive. The presence of living macroinvertebrates in filters in water treatment plants has been reported worldwide. Copepoda, Rotatoria, and Diptera are the most common taxa in water treatment plants [9,12,13,16,17,18]. However, most previous studies have confirmed the presence of invertebrates in filters, and few studies have been conducted on the possibility of physical filtration of larvae in water treatment plants, along with their ecological aspects. By identifying that possibility, we can determine whether current water quality standards are adequate or any further steps may be required.

The objective of this study was to assess the passability of chironomid larvae in GAC using *Glyptotendipes tokunagai*, one of the most dominant species in urban streams in Korea, at the laboratory scale. We tested the passability of larvae in the GAC at larval instar stages and interpreted the results ecologically. In addition, the time required for larvae to pass through the GAC column was verified.

## 2. Materials and Methods

### 2.1. Experimental Species

The species used in this experiment, *G. tokunagai*, is distributed throughout East Asia and was first recorded in Korea in 1981 [7,19]. They are one of the dominant species in urban streams because of their pollutant tolerance, and they are used for various laboratory experiments because they are easy to rear indoors [7,20,21,22]. *G. tokunagai* has four larval stages [7]; accordingly, in this experiment, the passability of larvae in the GAC was tested for each larval stage. The larvae used in our experiment were supplied by the laboratory at Division of Environmental Science and Ecological Engineering, Korea University (Seoul, Korea), where *G. tokunagai* was reared indoors. Egg masses were oviposited at almost the same time, reared at a constant temperature (approximately 25 °C) in the laboratory, and each time the larvae hatched or molted at each stage, they were used in the experiment as soon as possible. Before the injection of larvae, we confirmed the head width of larvae using stereo microscope (SZ61, Olympus, Tokyo, Japan) to double check the instar stages [23].

### 2.2. Experimental Procedures

To test the passability of *G. tokunagai* larvae in GAC used in water treatment plants, acrylic circular columns filled with GAC were used in the experiment (Figure 1a). The standard size of the columns was 300 mm in height and 27.5 mm in diameter. In each column, 1 cm of sand (particles approximately 2 mm in diameter) was laid on the bottom, and 23 cm of GAC (Kaya Activated Carbon Inc., Seoul, Korea) was placed at the top. The ranges of particle size and uniformity coefficient (UC) of GAC in Criteria for Waterworks Facilities [4] are 0.4–2.4 mm and 1.3–2.1, respectively. According to these criteria, the ranges of particle size and UC of GAC used in this experiment were 0.6–2.3 mm and 1.7, respectively. Thirty *G. tokunagai* individuals within the same larval stage were injected, and this step was repeated for each of the four larval stages without food resources. Because the standard filtration speed of the rapid filter system was 120–150 m·d^−1^ [4], the filtration speed in each column was maintained at approximately 150 m·d^−1^ (flow rate 63 mL·min^−1^) with the use of a submersible pump (UP 500, Hyubshin Water Design, Seoul, Korea) according to the standard. The linear velocity was 6.4 m·h^−1^, which met the criteria for waterworks facilities (2–34 m·h^−1^) [4]. Tap water was used for the experiment, and the average turbidity, residual chlorine, pH, and temperature of the water were 0.03 NTU, 0.94 mg·L^−1^, 7.09, and 25.2 °C, respectively. The larvae were screened from the water filtered by the GAC using a 45-µm sieve (ChungGye Sieve, Seoul, Korea) (Figure 1b). Because the recorded minimum head width of the first instar larvae was approximately 0.04 mm [23], the first and second instar larvae were double-screened by adding a 25-µm sieve. The number of passing individuals was recorded every 8 h after the larvae were injected. At each confirmation, a new sieve was placed on the outlet tube, and the larvae were visually confirmed using a stereo microscope (SZ61, Olympus, Tokyo, Japan) as they were in the filtered sieve for 8 h. The experiment was conducted for 168 h, because the average backwashing cycle in domestic water treatment plants was 7 d. The average development time of *G. tokunagai* larvae stages with sufficient food supply is approximately 147 h (at 25 °C), and since the development period of chironomid becomes longer under resource-limited conditions, we assumed that there was no change in the larval stage at 168 h of experiment [7,24]. To determine the location of the larvae remaining in the GAC layer after 168 h, the GAC was poured into a white tray (1426B Larval tray, BioQuip, Rancho Dominguez, CA, USA), and the number of larvae was counted for every 3-cm section of the GAC material.

### 2.3. Data Analysis

The recommended *t*-test for small sample sizes [25] was used to compare the density of *G. tokunagai* larvae in the GAC columns according to the instar. The Kruskal-Wallis test, followed by Bonferroni’s multiple comparisons, was used to test for differences in larval densities according to depth. The Jonckheere–Terpstra test was used to determine the depth dependency. All statistical analyses were performed using SPSS version 25.0 at a significance level of 0.05.

## 3. Results and Discussion

With respect to the passability of larvae in the GAC according to larval stage, only first and second instar larvae passed through the GAC, and the pass rate of first instar larvae was higher than that of second instar larvae (*t*-test, *p* = 0.01). First instar larvae were first detected 48 h after being injected into the GAC column, and more than 80% of the injected larvae passed after 72 h. In the case of the second instar larvae, passing individuals were first confirmed 72 h after injection. There were no passing individuals in the third and fourth instars (Figure 2). Because backwashing of GAC has a great effect on water quality in water treatment plants, backwashing conditions and cycles are very important factors to consider for efficient water quality management [26]. The backwashing cycle of activated carbon filters in domestic water treatment plants occur once every 3–5 days during the summer season [27]. According to our experimental results, the backwashing cycle may need to be adjusted to less than 48 h to prevent the chironomid larvae from spreading to the drinking water distribution system during the activity period of chironomids. Optimization of the backwashing procedure is known to be efficient in the removal of invertebrates [18], and our results can be helpful in determining the backwashing cycle of water treatment plants.

At the end of the experimental period (168 h), we confirmed the location of the larvae in the GAC columns in 3 cm sections. The first instar larvae were excluded from this test because more than 80% of the injected individuals passed through the GAC. The percentage of larvae injected to passing larvae was 63.3% in the second instar, 65.6% in the third instar, and 44.4% in the fourth instar. The loss of larvae was believed to be due to cannibalism resulting from a lack of food in the experimental environment [28]. Organisms generally require more energy as they grow; therefore, fourth instar larvae may have higher energy requirements than larvae at earlier stages, resulting in more cannibalism in the fourth instar.

Second instar larvae were evenly distributed throughout GAC (Figure 3a). At the 0–3 cm section, the densities of third and fourth instar larvae decreased significantly with increasing depth of GAC (Jonckheere–Terpstra; Figure 3 and Table 1). Among the confirmed third instar larvae, 86.4% were present at a depth of 3 cm from the inlet, and 94.9% were present at a depth of 9 cm (Figure 3b). In the case of confirmed fourth instar larvae, they were mostly found below the inlet, with 95.0% distribution within 3 cm (Figure 3c). Oliver [29] reported that approximately 95% of chironomid larvae usually burrow within the upper 10 cm of the substrate, and our results also reflected this larval habit. Larvae construct tubes composed of endogenously produced silk and substrate materials [30,31], and the third and fourth instar larvae seem to have built a tube and withstood the flow of water rather than passing through the GAC. Newly hatched benthic macroinvertebrates use flowing water to disperse from hatching sites to appropriated habitats, and chironomid larvae also float and remain in a planktonic state until they find a suitable habitat [29,32]. This is likely the reason why over 80% of the first instar larvae passed through the experimental GAC in our study.

When macroinvertebrates can be seen in domestic tap water, their aesthetic impact on perceived water quality is enormous, even if the frequency of such occurrences is low. In studies of benthic macroinvertebrates in drinking water systems in Europe, not only chironomid larvae, but also various macroinvertebrates such as water louse (*Asellus aquaticus*), cave water louse (*Proasellus cavaticus*), and detritus worms (*Oligochaeta* spp.) were found [11,33]. However, in these studies, samples were collected from water reservoirs or pipes in drinking water supply networks [11,14,34]. In the current study, we confirmed the possibility of macroinvertebrates physically filtered from the GAC used in a water treatment facility, which is a stage before pipe inflow. If chironomid larvae pass through GAC, they may enter the drinking water networks, where they might reproduce within the pipes and eventually enter the home water supply. A few species of Chironomidae, such as *Limnophyes asquamatus* and *Paratanytarsus grimmii*, which are widely distributed in North America, Australia, and Europe, can multiply parthenogenetically without reaching the adult stage [11,35]. Egg masses of Chironomidae have been found to serve as a reservoir for Vibrio cholera and *Aeromonas* spp., and Chironomidae may cause the spread of pathogenic bacteria into drinking water networks [36].

Maintaining good water quality requires various efforts, such as the management of water treatment processes (including regular pipe flushing and backwashing cycles) and monitoring the occurrence of macroinvertebrates. Our experiment confirmed the passability of chironomid larvae in GAC at different instars, and can provide guidelines for estimating water treatment standards. The first instar larvae passed through the GAC after 48 h, and this could be evidence suggesting that the backwashing cycle of water treatment plants should be controlled to within 2 d. Moreover, it is expected to provide basic data for improved experiments such as the testing of GAC filtering efficiency with different variables or the development of a GAC sorption model.

## 4. Conclusions

The supply of clean water is often taken for granted; thus, the detection of chironomid larvae in tap water is a cause of anxiety for the general public. In response to civil complaints, the Korean government has proposed management plans to provide high-quality water. With this goal in mind, we conducted experiments on the physical passability of chironomid larvae in GAC used in water treatment plants, and the results were interpreted in relation to the ecological characteristics of Chironomidae. Our experimental results demonstrated the potential for chironomid larvae to be transmitted into drinking water networks using the current management method, and the data can be used as a reference for establishing guidelines for the management of water treatment plants. Currently, the backwashing cycle of activated carbon in domestic water treatment plants is three to five days in summer, but we propose to adjust the cycle to less than 48 h during the activity period of the chironomid. Improved studies are needed to relieve consumers’ anxiety about the occurrence of invertebrates in tap water, and our study can be used as basic data for those studies.

## Figures and Tables

**Figure 1 ijerph-19-01005-f001:**
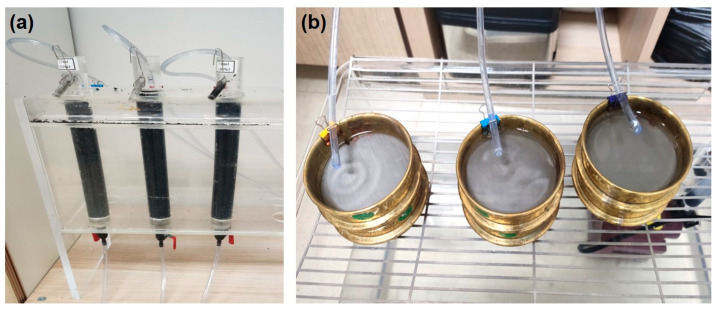
The columns filled with granular activated carbon (GAC) used in the experiments (**a**). Larvae that passed through GAC column were confirmed by filtering using sieves (**b**).

**Figure 2 ijerph-19-01005-f002:**
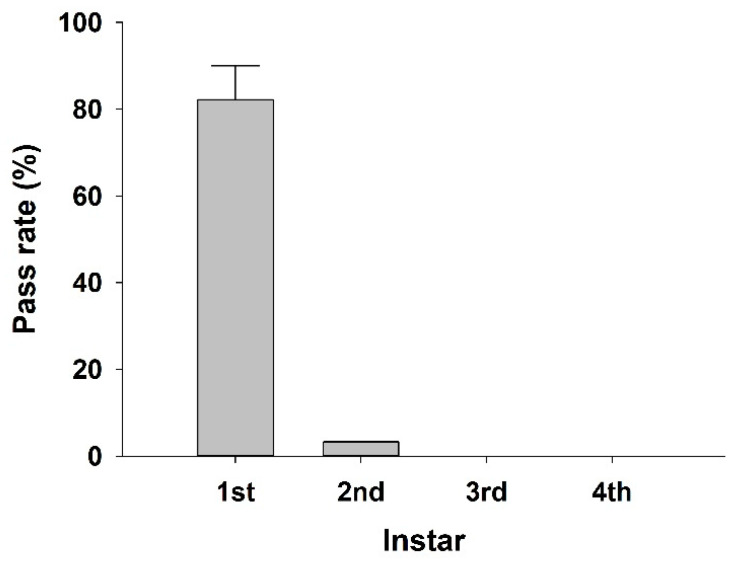
Pass rate of the four instars of *Glyptotendipes tokunagai* larvae through a granular activated carbon (GAC) column.

**Figure 3 ijerph-19-01005-f003:**
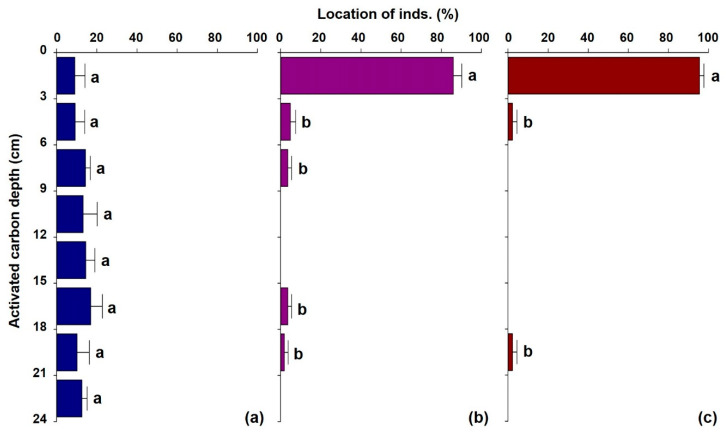
Distribution ratio of *Glyptotendipes tokunagai* larvae detected in 3-cm sections of GAC after 168 h of passage by larvae in the second instar (**a**), third instar (**b**), and fourth instar (**c**). Different letters indicate statistically significant differences at the 0.05 level in the Jonckheere–Terpstra test.

**Table 1 ijerph-19-01005-t001:** Results of Kruskal–Wallis and Jonckheere–Terpsta tests determining whether significant differences exist between the abundance of larvae at different depths in the granular activated carbon (GAC) column.

Instar	Kruskal-Wallis	Jonckheere–Terpstra
Second	NS ^1^	NS
Third	NS	0.009
Fourth	0.05	0.034

^1^ NS = not significant.

## Data Availability

The data presented in this study are available on request from the corresponding author.

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
