# Peer review of "Passability of Chironomid Larvae in Granular Activated Carbon"

_ijerph, 2022, doi:10.3390/ijerph19021005_

Round 1

Reviewer 1 Report

The manuscript provide an applicable and high realizable way to reduce benthic macroinvertebrates level at the initial stage of water treatment. However, the results can be discussed more deeply. For example, some sorption model such as Langmuir adsorption model can be applied based on the present experiments with different concentration of larvae. Moreover, characteristics of the GAC such as SEM photos may be introduced into the manuscript. The reuse time of GAC can also be determined. 

Author Response

The manuscript provide an applicable and high realizable way to reduce benthic macroinvertebrates level at the initial stage of water treatment.

Answer: We are grateful to the reviewer for the insightful comments on the manuscript. We have made changes to reflect most of the suggestions provided by the reviewer. We have marked the revisions with 'track changes' in the manuscript. Here is a point-by-point response to the reviewer’s comments and concerns, and all line numbers refer to the revised manuscript file with tracked changes.

However, the results can be discussed more deeply. For example, some sorption model such as Langmuir adsorption model can be applied based on the present experiments with different concentration of larvae.

Answer: Thank you for your suggestion. Our experiment is a preliminary study to develop sorption models as you suggested, and it is expected to provide basic information related to it. Therefore, we added a comment on the applicability of the results of this experiment (lines 195-197).

Moreover, characteristics of the GAC such as SEM photos may be introduced into the manuscript. 

Answer: Thank you for your suggestion. We have added an explanation of the characteristics of the GAC in the manuscript (lines 88-91).

The reuse time of GAC can also be determined.

Answer: We proposed the adjustment of the backwashing cycle of the GAC in the water treatment plants (lines 133-138), and described it more clearly in the manuscript (lines 191-195).

Reviewer 2 Report

This paper evaluates the passability of chironomid larvae in granular activated carbon. In my opinion, the data are not enough to publish as a paper. Also, the novelty is not clear why this study is important and should be published. My suggestion is rejection. 

Line 55: the cited studies are not acceptable to show the novelty of this paper (two of them are too old).

You should have a more strong reason why the hypothesis of this study is important. Why the possibility of larvae in the GAC is important? You should support your idea by using strong updated references. 

Author Response

This paper evaluates the passability of chironomid larvae in granular activated carbon. In my opinion, the data are not enough to publish as a paper. Also, the novelty is not clear why this study is important and should be published. My suggestion is rejection.

Line 55: the cited studies are not acceptable to show the novelty of this paper (two of them are too old).

You should have a more strong reason why the hypothesis of this study is important. Why the possibility of larvae in the GAC is important? You should support your idea by using strong updated references.

Answer: Thank you for your suggestion. We added a description to explain the difference between the previous studies and our study, and the usefulness of the novelty (lines 45-52, 55-59). We have also added updated references as suggested (lines 52-55). 

Reviewer 3 Report

The publication takes up an important topic, but the presented solution is unclear. There is also no influence of various factors in the water on the retention of larvae. eg pH, other ions, temperature, flow rate, time. an effective research method should also be selected. In the resulting part, apart from the indicated aspects of variables, samples of the visualization of the deposit should be added in consecutive centimeters of column length.

Conclusions should relate to the results of the research, including long-term conclusions

Author Response

The publication takes up an important topic, but the presented solution is unclear. There is also no influence of various factors in the water on the retention of larvae. eg pH, other ions, temperature, flow rate, time. an effective research method should also be selected. In the resulting part, apart from the indicated aspects of variables, samples of the visualization of the deposit should be added in consecutive centimeters of column length.

Answer: Thank you for your suggestion. Our experiment was conducted to confirm the passability of chironomid larvae in granular activated carbon under conditions similar to those of general water treatment plants and to provide basic information to help establish waterworks management. Therefore, we added descriptions of the experimental factors related to these conditions in the manuscript (lines 88-100).

Conclusions should relate to the results of this research, including long-term conclusions.

Answer: Thank you for your suggestion. We have corrected the conclusions in relation to our experimental results (lines 208-212).

Reviewer 4 Report

The authors investigated the passability of chironomid larvae in granular activated carbonin drinking water treatment. The topic is urgent and interesting. The manuscript can be published after revisions as follows.

1) The performance of GAC would be determined by the diameter of GAC and linear velocity inside the column. The details (diameter, manufacturer, sieving properties, and so on) of GAC used in the experiments must be given. The linear velocity (feed water volumetric rate divided by cross-section area of the column) must be given.

2) In drinking water treatment, as the authors mentioned, chironomid larvae could be removed by sand filtration in the case of the treatment for surface water. It is necessary to mention about the target treatment process (coagulation - sedimentation - GAC - sand filtration, coagulation - sedimentation - filtration - GAC or other) which authors intended to investigate. In addition, some readers misunderstand that the GAC can be designed to remove chironomid larvae. The authors are advised to mention the objective of this study more clearly to avoid misunderstandings.

3) Chironomid larvae could migrate slowly inside the column. The reason for the operational period of 168 h must be explained.

4) The volume of water used for the examination of chironomid larvae must be mentioned, when the authors examined their presence every 8 h.

5) The conclusion must be more quantitatively mentioned.

6) The methods for preparation of chironomid larvae in each instar stage. The possibility of change of the stage during 168 h of experiments inside the column has to be mentioned.

Author Response

The authors investigated the passability of chironomid larvae in granular activated carbon in drinking water treatment. The topic is urgent and interesting. The manuscript can be published after revisions as follows.

Answer: We are grateful to the reviewer for the insightful comments on the manuscript. We have made changes to reflect most of the suggestions provided by the reviewer. We have marked the revisions with 'track changes' in the manuscript. Here is a point-by-point response to the reviewer’s comments and concerns, and all line numbers refer to the revised manuscript file with tracked changes.

1) The performance of GAC would be determined by the diameter of GAC and linear velocity inside the column. The details (diameter, manufacturer, sieving properties, and so on) of GAC used in the experiments must be given. The linear velocity (feed water volumetric rate divided by cross-section area of the column) must be given.

Answer: Thank you for pointing this out. We agree with the reviewer's suggestion and have added detailed information on GAC and linear velocity in the manuscript (lines 88-91, 97-98).

2) In drinking water treatment, as the authors mentioned, chironomid larvae could be removed by sand filtration in the case of the treatment for surface water. It is necessary to mention about the target treatment process (coagulation - sedimentation - GAC - sand filtration, coagulation - sedimentation - filtration - GAC or other) which authors intended to investigate. In addition, some readers misunderstand that the GAC can be designed to remove chironomid larvae. The authors are advised to mention the objective of this study more clearly to avoid misunderstandings.

Answer: Thank you for these comments. We added a description of the purification process of water in the water treatment plant (lines 29-33) and described the purpose of our study (lines 55-59).

3) Chironomid larvae could migrate slowly inside the column. The reason for the operational period of 168 h must be explained.

Answer: Thank you for your suggestion. We have added a description of the reason for the operational period of 168 h (lines 107-108).

4) The volume of water used for the examination of chironomid larvae must be mentioned, when the authors examined their presence every 8 h.

Answer: Thank you for your suggestion. Unfortunately, it was more difficult to identify when the larvae were transferred from the sieve to a tray with water. Therefore, whenever we checked the larvae every 8 h, we placed a new sieve on the outlet tube and checked the presence of larvae using a microscope as they were in the sieve. This method is easier and less time-consuming. We have added a detailed description to the manuscript (lines 104-106).

5) The conclusion must be more quantitatively mentioned.

Answer: We modified the conclusions to include our experimental results quantitatively, as suggested (lines 208-212).

6) The methods for preparation of chironomid larvae in each instar stage. The possibility of change of the stage during 168 h of experiments inside the column has to be mentioned.

Answer: Thank you for these comments. We have added a detailed description in the Materials and Methods (lines 72-77, 108-111).

Round 2

Reviewer 2 Report

The authors tried to show the novelty of this research by adding lines 52-70. But, after reading I could not find a related and recent study that shows the gap in the field. The cited studies 16-18 are too old and not acceptable as a research gap. I expect to see other studies performed in Korea. Maybe the results are good, but not enough for the publication in IJERPH. Unfortunately, my suggestion is rejection. 

Reviewer 3 Report

After introducing the necessary modifications, the publication improved its quality. Still this topic should be thoroughly analyzed but it can also be published in this form. The scientific level increased after the development of the experimental part, but still does not exceed the average level. The topic will also be of interest to a small group of readers.